# Manufacturing Bacteriophages (Part 1 of 2): Cell Line Development, Upstream, and Downstream Considerations

**DOI:** 10.3390/ph14090934

**Published:** 2021-09-17

**Authors:** Tayfun Tanir, Marvin Orellana, Aster Escalante, Carolina Moraes de Souza, Michael S. Koeris

**Affiliations:** Amgen Bioprocessing Center, Department of Biological Engineering and Management, Keck Graduate Institute, Henry E. Riggs School of Applied Life Sciences, Claremont, CA 91711, USA; TTANIR20@students.kgi.edu (T.T.); MORELLANA20@students.kgi.edu (M.O.); ASTER_ESCALANTE@kgi.edu (A.E.); CMORAESDESO18@students.kgi.edu (C.M.d.S.)

**Keywords:** manufacturing, bacteriophage, cell line development, upstream processing, downstream processing, drug substance, drug product

## Abstract

Within this first part of the two-part series on phage manufacturing, we will give an overview of the process leading to bacteriophages as a drug substance, before covering the formulation into a drug product in the second part. The principal goal is to provide the reader with a comprehensive framework of the challenges and opportunities that present themselves when developing manufacturing processes for bacteriophage-based products. We will examine cell line development for manufacture, upstream and downstream processes, while also covering the additional opportunities that engineered bacteriophages present.

## 1. Introduction

As bacteriophage therapy progresses in clinical trials towards an initial FDA and potentially EMA approval, more researchers and developers are interested in the necessary steps to move from a potential therapeutically useful phage or phages to a drug product (DP) that can be tested clinically and eventually approved. The manufacture of phages initially leads them to be drug substance (DS), and we focus this first part of the two-part series on phage manufacturing on the processes leading up to and including the final DS. We will be providing an overview of the processes and considerations for the aspiring bacteriophage drug developer as it regards choices, challenges, and tradeoffs to be made. The canonical steps include design and creation of a master cell bank and a master phage bank in cell line development (CLD), followed by production of phages in their cognate hosts during upstream processing (USP), and ending with pure DS after downstream processing (DSP), i.e., all purification steps necessary. The DS is then formulated and filled, which we cover in the second part of this series.

The principal goal is to provide the reader with a comprehensive framework of the challenges and opportunities that present themselves when developing manufacturing processes for bacteriophage-based products, while considering unmodified and modified or engineered bacteriophage and bacterial cell lines.

For those readers interested in the regulatory aspects of bacteriophages we reference them to the review in this Special Issue covering the recent FDA/NIAID bacteriophage regulatory workshop and implications thereof.

## 2. Cell Line Development (CLD)

We begin by defining the steps for any and all bacteriophage-based products to go through in order to be approvable by at least FDA and EMA. This includes all phages that are part of the therapeutic cocktail, as well as all productions hosts need to be developed and banked, which need to follow the guidelines from the FDA and EMA related to regulation and operations of biological drugs. These guidelines are set for the raw materials to be prepared and characterized as they become part of a cell bank, and in this case also a phage bank [1]. This example is analogous to the creation, validation, and operation of mammalian viral product and the viral and cell banks [2]. In addition, the FDA and EMA state that the cells being used to make master cell banks must pass set quality standards for identity, potency, purity, and as part of cell characterization [3]. Methods for testing and meeting these requirements will be discussed further in the second part of this series.

### 2.1. Manufacturing Inclusion Criteria for Candidate Phages

During the initial selection, specific attention must be paid to upstream yield to satisfy inclusion criteria. It is critical to know how many phages are needed for an efficacious therapy, yet the preference from the process development (PD) and CLD side is for minimizing the total number of phages, and the total number of production hosts needed, while maintaining clinical efficacy. The combination of phage and production host further drives choices as yield discrepancies between phage-host combinations within the cocktail are a significant risk for PD success and overall manufacturing. To be clear, it is critical to know the potency of the phage DS, which, on the upstream side, is initially a product of lysis time and burst size. For these types of experiments, different phage concentrations should be tested on a given host concentration and lysis timing should be measured. A measure of phage fitness can be described as the ratio of phages to bacteria generated per unit time [4]. The time at which the highest ratio is achieved would be considered the maximum fitness of that phage for the system tested in a well-mixed liquid environment [5]. Although it might seem intuitive that the maximum fitness is achieved by phages whose lysis time is longer, which leads to more phages produced per cell, there exists an optimum lysis time beyond which new infection is preferred. In the presence of a high host concentration, lysis time can be shortened as phages bias themselves to infect more host cells [6]. Consequently, the shorter lysis time and greater infection rate leads to a higher phage to bacteria ratio; a better phage fitness is achieved. We encourage everyone to carefully explore, measure, and calculate optimal best burst size and lysis time combination for the phage [6]. The shortened lysis time is due to the available resources in the environment that allow for a larger phage population [7]. Another factor to consider before introducing a phage to the host is the host density. Some phages only show replication at a certain host density, so it is important to know the density required before phages are introduced in a bioreactor [8]. In addition, it is worth noting that the host density required or favorable for phage growth is different from multiplicity of infection in one important aspect. Host density relates the number of bacterial target cells to themselves per unit volume, which is an important parameter as it informs the interested drug developer about the physiological state of the target bacteria. Both density and MOI are critical to characterize. In addition to host density, host growth rate also plays a role in phage lysis time and burst size. It has been shown that bacteria with a higher growth rate induce a higher burst size in phages and host replication at the maximum growth rate induces the highest burst size [7,9]. Therefore, after determination of the host density for growth, the maximum growth rate can be determined if a large burst size is desired.

### 2.2. Purity and Identity

Once phages and hosts have been selected, certain purity requirements need to be met. This is well verifiable for the phages through various physiochemical, microbiological, molecular, and genetic analytical methods, but special care must be taken for the microbial hosts. All bacterial production hosts should be tested and be free of contaminating phages. This includes infection with a contaminating phage that is not the dominant infecting phage, as well as a lysogen or prophages. Detection of prophages can be performed using common methods such as stress-based induction (irradiation or chemical [10,11,12,13]), as well as through bioinformatic methods [14,15,16]. For less sensitive viral detection or known set of viruses being tested, a simple PCR method can be used for detection [17]. Protein-based methods such as ELISA and Western blots can also be used to detect specific phage antigens that could potentially contaminate a sample. Lastly, and especially for engineered bacteriophages, it is critical to determine the purity and identity of the master phage bank sample in order to determine if any mutations occurred. This topic will be explored in more detail in the upstream processing section.

### 2.3. Master Phage Bank Design

When constructing a master phage bank (MPB), it is important to understand phage characteristics, such as MOI and related them to the requirements of the process including scale and number of batches to be produced. To properly size a MPB it is crucial to be able to accurately measure phage concentration and viability in each step.

A popular, but hard-to-control, method for determining phage count is known as the double-layer plaque assay, which is a simple and affordable technique that can be applied in most labs and is considered acceptable in the field for the purpose of phage enumeration [18]. This method relies on the ability of phages to lyse bacteria and form visible plaques that are established with continuous serial dilutions to then determine the plaque forming unit (PFU) [19]. Although this is a historical method for phage enumeration, updated and more precise methods are available and emerging into the field for this purpose of determining phage titer [19,20,21,22]. We strongly urge anyone looking to quantitate phages to evaluate several methods and consider assay complexity, robustness, and repeatability when making a choice about how to assess phage concentration and viability. Failing to ensure a robust analytical capability leads at best to rejecting batches during manufacture, and at worst (for now) failure in clinical trials. We further discuss analytical considerations in part 2 of this review series.

### 2.4. Storage

Having isolated the appropriate number and type of phages and production hosts for the manufacture of a therapeutic cocktail, we now turn to optimal long-term storage. We focus here on the storage of phages, as the long-term storage of bacteria presents fewer challenges. Phage preservation begins at refrigeration temperatures and moves down from there. For the purposes of this section, we do not consider freeze- or spray-drying for master phage banks due to the considerable development time involved. For a short-term storage 4 °C has shown to be sufficient for 12 months of storage with common excipients for purified phages or even with raw lysate [23]. Although this report only claims one year of stability, in another report the addition of 50% *v*/*v* of glycerol allowed storage of up to 2 years [24]. Similarly, periods of more than a year at 4 °C have been reported with a multi-phage preparation [25]. However, this is not a uniform behavior, nor does 4 °C sufficiently protect phages from degradation [26]. On the other end of the spectrum we have the extremely deep freezing storage in liquid nitrogen which showed no appreciable degradation for a 5 year period [24]. However, that type of storage might be impractical for broad use. To strike a balance most process development and manufacturing facilities are equipped with −80 °C freezers, and one group reported storage data of over 400 bacteriophages for more than 10 years, also reporting that some phages did not fare well at deep-freezing conditions [27].

If purified phages are difficult to store another method to consider—but that has not been considered by the FDA or EMA yet—is storage of phage with the bacterial host. This process includes introduction of phages to the bacteria and culturing for a short period of time. A sample of the phage and host medium is taken and mixed with a volume of glycerol to make a 15% glycerol mix [28]. The use of phage storage within the bacterial host has also been suggested for use with phages that show recover poorly from cold storage [29]. This method can be used for development purposes for the storage of new phages with unknown storage properties.

In addition to phage lysate and whole host and phage storage, other methods have been suggested such as lyophilization [30]. However, this method is generally not preferred due to the large loss of titer and development time [24]. If the process developer intends to invest in this method however, the extensive screening of excipients during lyophilization, such as sucrose, glucose, and gelatin, alone or in combination, amongst others is recommended [31].

### 2.5. Phage Engineering Opportunities and Challenges in Cell Line Development

During development, phages can be engineered to have enhanced properties that will allow them to better infect their host, while overcoming challenges, such as phage resistance, to name just a few [32]. One problem phages encounter with bacteria are the various host-defense system [33]. This limits their clinical efficacy but also limits their choice of production host and should be considered from that perspective as well. Recoding of the phage genome might be one approach, or recently discovered anti-CRISPR proteins can be leveraged [34]. More and more anti-CRISPR proteins are being discovered, enhancing the available toolset [35]. Another phage engineering approach common in the evaluation of therapeutic utility is mutagenesis or engineering of phages within the tail fiber region. The generation of phages with mutations in these regions had an increase in host range and allowed infection in previously resistant host [36]. This is of particular utility within the context of CLD as an expanded host range might lead to a reduction in the complexity or number of the required production hosts. This benefit cascades not just through CLD, into USP but also lessens the regulatory burden.

## 3. Upstream Processing (USP)

Upstream process development provides the necessary transition from the laboratory environment to a manufacturing environment where the target product is produced the desired quality target product profile (QTPP) [37,38,39]. Understanding the biology of the phage infectious cycle remains crucial in the design of upstream processes [40]. This cycle consists of adsorption of the phage to the host cell, injection of the genetic material (also called penetration), amplification of the phages and the lysis of the host. Phage replication within the context of manufacturing is driven by three main parameters that need to be considered: (i) adsorption constant (the rate at which phages attach to bacteria), (ii) latency time (the time between attachment to lysis), and (iii) burst size (the number of phages released from a bacterium) [41]. To optimize the phage titers, the process should be designed to minimize latent time and maximize adsorption rate and burst size. Host related factors such as physiological state of the host [42], its metabolic activity [43], specific growth rate [7], and the density of phage receptors on the host [44] affect the latent period, burst size, and adsorption rate. Further parameters that should be carefully considered in upstream processing (USP) which we will go into greater detail below are production modes, composition of growth media, bacterial growth, and infection temperatures, pH, MOI, initial bacteria concentration, time of infection, threshold density, and dilution rate in the case of continuous production. Moreover, the phage(s) or host strains to be used in the manufacturing should be free of virulent genes (antibiotic resistance, toxins, etc., to lessen the purification burden) and the culture conditions should be designed in a way to prevent coevolution of host and phage [45].

### 3.1. Phage Engineering Opportunities and Challenges in USP

Engineering phages can help in many aspects of both manufacturing of phage therapy and its efficiency against the target organism [46]. However, once the engineering is completed, one must guard against the accumulation of spontaneous mutations. The more the phages replicate the greater the chance of acquiring spontaneous mutations. This requires limiting the duration of the process to a certain time to avoid high amounts of mutations to accumulate. Therefore, the replicase of the phage of interest may be substituted with a higher fidelity replicase which could help reducing the spontaneous mutation rate. Replication deficient phages may offer some benefits, such as ensuring the engineered-phages not to be released to the nature, and to be able to calculate the required doses precisely [32,47]. Whilst the aim of making lysis-incompetent phages were to avoid toxin release in patients’ body, immune response against phages, and the host to develop immunity against phages, this can be also useful in manufacturing. Utilizing this approach, the amount of toxins released during the upstream process can be reduced as bacterial lysis is avoided which will not only ease downstream processing but also reduce the chance of formation of resistant host strains during the production process [47].

### 3.2. Critical Process Parameters for USP Phage Production

#### 3.2.1. The Effect of Temperature on Phage Production

Temperature remains one of the parameters that are important for phage production as it affects the attachment of phage to its host, penetration of phage genetic material, its amplification, and the latent time: low temperatures decrease penetration of phage genetic material and high temperatures may lead longer latency [48]. Many studies suggest using the same temperatures for bacterial growth and infection, but aside from convenience there is no other reason that suggests not exploring combinations of temperatures and shifts. One historical study by Pollard et al. demonstrated using different temperatures to obtain differential titers [49]. Another example is the lowering of the infection temperature for *E. coli*—T4 [50,51]. Grieco et al. reported that in a Design of Experiment study where the effect of DO, pH, and temperature was tested for optimal yield, the optimal theoretical yield (2.86 × 10^11^ PFU/mL) was achieved at 28.1 °C and a pH of 6.9 which was validated with empirical studies (3.49 × 10^11^ PFU/mL). A more extreme example of the benefit of temperature reduction during infection is shown with the success of moving from 37 °C to 25 °C, yielding obtaining higher titers [52]. However, as with most optimization problems pertaining to phages, some studies suggest the opposite: in a study by Zaburlin et al. in 2017, latent and burst times are shown to be prolonged at low temperatures (20–25 °C) and at low pH (5–6) [13]. Furthermore, some bacteria (some strains of *Listeria monocytogenes*) reported to activate their restriction modification systems that enable them to develop resistance against phages at ≤30 °C [53]. However, some phages might be able to overcome that particular host-defense system where the loss of the ability of phages LP-048 and LP-125 to infect *L. monocytogenes* strains at 37 °C is compensated by their ability to infect at 30 °C [54]. All of these examples strongly suggest designing manufacturing systems that can accommodate variations in conditions, as well as feeding optimal parameters into the choice algorithm for cocktail composition.

#### 3.2.2. The Effect of Media Composition on Phage Production

The composition of bacterial growth media is one of the important parameters to produce phages as it is reported to affect phage adsorption and bacterial physiological state [55]. Complex media provide better bacterial growth compared to chemically defined media (CDM); however, it includes organic compounds that may introduce contaminants and also prolonging the time required for the separation and purification. Furthermore, yeast extract is reported to hamper phage amplification among the organic ingredients [55]. Since safety is crucial in terms of phage therapy, high bacterial biomass production can be sacrificed to a certain extent for the sake of safety and quality of the product. Additionally, studies focusing on developing more efficient synthetic media to obtain higher biomass show good progress: the two CDMs (ZMB1 and ZMB2) provided higher bacterial biomass than a commonly utilized complex media (M17) [56]. In another study, the researchers developed a new defined media called SM-1 that has free amino acids as the sole nitrogen source to avoid using large organic molecules and to reduce the length of the lag phase which are important parameters in industrial scale production [55]. In the future chemically defined media might become the default choice for developers of phage therapies.

#### 3.2.3. Multiplicity of Infection (MOI)

MOI refers to the number of phages per host cell and is a very important parameter to determine for specific host-phage systems to optimize phage yields. Interaction between bacteria and phages mostly governed by concentration. Thus, optimal concentration is of great importance to provide maximum infection and replication. The choice of MOI is critical for productivity as well. An optimal MOI should allow a certain number of cells to remain uninfected and continue to grow to produce more uninfected cells to increase productivity. Moreover, it is critical to determine the optimal MOI for phage-host system to determine the required phage inoculum for large scale production [57,58]. In addition to MOI it was thought that there was a threshold necessary to be crossed to have bacterial hosts be susceptible to and productive for phage amplification to occur [8,59].

In the production of phages for medicinal use, the genetic stability of host and the host-specific phage remains crucial. Because, as the host bacteria undergoes spontaneous mutations throughout the production process, they may become resistant to phage of interest or the phage may differ from the desired final phage structure that leads changes in the function, productivity, and yield [60]. Therefore, the spontaneous mutation rates of both host bacteria and phage(s) is of paramount importance. Phages with dsDNA genomes have higher spontaneous mutation rates compared to their hosts (as measured for medium and large bacteriophages between 10^−7^ and 10^−9^ substitutions per nucleotide per cell infection (s/n/c) [61]), which translates to a general need to minimize the number of infection cycles while still maximizing yield [62,63]. Due to the increasing awareness for antibiotic-resistance among bacteria and preservation of microbiota, large quantities of phage production for medicinal use remains crucial [64]. Therefore, an optimal number of phage replication cycles should be determined for the phage of interest and the production should be designed considering this. Since each phage has a different replicase fidelity, the number of replication cycles without significant change in the phage genome for each phage will differ.

#### 3.2.4. Production Modes for Phage Manufacturing

We distinguish between the three main production modes: batch, semi-continuous, and fully continuous (Table 1).

##### Batch Process

In this mode of production, phage infection occurs in the same fermentor where bacterial growth occurs. Since the process is in batch mode, the total working volume is limited to the largest equipment size available although manufacturing of high titters of phages are reported with batch production. Batch process is the cheapest mode of manufacturing to produce phages [58], and it is capable to achieve high titers in some configurations [55,66,68]. Batch processes do have some downsides, such as potentially high downtime to production time ratio, high cost of capital investment, and lot-to-lot product quality variability [67]. The process flow diagram (PFD) in Figure 1 shows a generalized batch phage production scheme. The general batch process starts by feeding the media into the fermentor and following the inoculation and amplification of the host-bacteria. As the bacteria enters exponential phase, the tank is inoculated with the optimal MOI of phages and harvested after lysis occurs.

##### Semi-Continuous Process

Semi-continuous processes (or SCF/SCI—self-cycling fermentation/self-cycling infection process) proposed by Sauvageau and Cooper (2010) to combine the advantages of both batch and continuous systems whilst improving the issues encountered in both modes of production [69]. The proposed method includes two separate fermentors for bacteria amplification and phage production (Figure 2). The fermentor where bacteria are grown is operated in cycles of batch production of bacteria (SCF) and half of the bacteria transferred to the phage amplification fermentor just before the growth phase reaches stationary phase and the transferred volume in the first tank is replaced with fresh medium. Phage amplification occurs in a separate fermentor that is operated in cycles of phage infection of bacteria (SCI). As each Infection cycle is completed, the culture medium in the SCF fermentor is harvested till the lowest level sensor in the fermentor and this whole SCF/SCI cycle is perpetuated.

##### Continuous Process

Continuous process provides significant benefits for industrial scale production of biopharmaceuticals as it allows long runs of production of the biopharmaceutical and, thus, reducing the downtime and increasing equipment utilization. Moreover, the physiological state of the bacteria can be regulated by changing the dilution rate and thus improving productivity. However, long-term utilization of continuous production mode in chemostats shown to lead coevolution of the host-phage system due to prolonged coexistence of host and phage which should be avoided in phage production for medical purposes. Thus, a two-stage continuous system (cellstat) is developed to minimize the co-evolution risk during the continuous production of phages. Mancuso et al. used a system (depicted in the Figure 3) consisting of two continuous stirred tanks and a final holding tank to enable continuous high-titer production of phages with no or minimal coevolution [67]. The system enables control over the physiological state of the host bacteria to keep them in their exponential growth phase where they are susceptible to phage infection and optimal phage production. The two reactors differ in volume which allowed adjusting residence times separately for each bioreactor, whilst keeping the flow rate the same throughout the system. This allows to create longer residence times for bacterial cells yielding higher bacterial concentrations, while it also helps with the infection due to short residence times in the bioreactor, minimizing the risk of coevolution. The last tank in the system acts as a holding tank for the infected cells for completion of their lysis away from selection pressure.

##### Single-Use vs. Stainless Steel

Single-use systems (single-use technology—SUT) are increasingly utilized in the biomanufacturing due to the various advantages they provide, effectively setting the standard in recombinant protein production for therapeutic use [70]. For bacteriophages it is conceivable to integrate SUT, as most phage production processes do not require different performance than that needed for most microbial fermentation that is performed using SUT. Single use technologies eliminate the extensive cleaning and validation steps reducing the time required for preparing the fermentor and, thus, increasing the overall productivity, the chance of cross-contamination, required number of facilities for cleaning materials, and reducing the overall energy consumption. Last, but quite possibly most importantly, using SUT drastically reduces changeover time from one component of a phage cocktail to next, a critical consideration as most therapy candidates will contain more than one phage.

## 4. Downstream Processing (DSP)

Downstream processing is focused on reducing or eliminating impurities derived from the production of a biological product. It is important to impress on the reader that the crude lysate becomes a DS along the purification continuum and should be treated as such (see Figure 4 for a qualitative representation). Large quantities of bacteriophages are often required for applications including acute infectious disease treatment [71]. Following upstream activities for phage propagation, phage concentration in the broth may be at 10^11^–10^12^ PFU/mL, which is roughly equivalent to a few mg/L of protein. With the rupture of host cells as well comes a variety of molecules, such as proteins, lipids, and nucleic acids, which can pose a challenge for less robust purification methods, such as chromatography or filtration [72]. Phages being used for commercial purposes need to be free of contaminants and impurities. These include, but not limited to bacteria, mold, lipopolysaccharides and host cell debris [73,74]. Before pre-formulation procedures, the crude lysate must be clarified and the phages captured, concentrated, and polished [72]. These steps constitute the major unit operations for phages as they relate to biologics manufacturing and serve to isolate and concentrate the phages for further processing [72]. For an overview schematic refer to Figure 5.

For a generalized process, impurity removal can be categorized as process-specific, product-specific, and purification-specific [75]. Process-specific impurities are derived from cell culture reagents, as well as additives like antibiotics, serum, and nucleases [75]. Product-specific impurities in this case are phage-related—empty capsids, aggregates, and foreign viruses are just a few examples [75]. Finally, purification procedures, such as chromatography, can inadvertently impact the purity of the phage stock by way of extractables and leachables [75].

### 4.1. Lysate Pre-Treatment

Following cell lysis, the phages may be collected by adding buffer directly to the lysate. It appears that some researchers elect to use a chloroform treatment for lysate preparation prior to traditional clarification procedures [73,76]. This serves to isolate the phage, although it has the potential to inactivate filamentous and lipid-containing phages [77]. A common issue of crude lysate is the presence of nucleic acids, which increases the viscosity of the lysate in proportion to the biomass concentration [72].

### 4.2. Crude Lysate Treatments

Not all purification methods are robust enough to tolerate cell lysate from harvest. Crude lysate may be tolerated by centrifugation, precipitation, and density gradient separation, but the impact of unprocessed lysate on chromatography and filtration would be detrimental to the performance of the equipment. Some of the most common methods for purification are simple, cost-efficient, and well-characterized. These include polyethylene glycol precipitation, cesium chloride gradient centrifugation, and filtration [74]. It has been reported that some phages do not respond well to centrifugal forces nor the interaction with cesium chloride—in the worst cases, the phages may be inactivated by the mineral salt [76]. Another consideration for purification includes the sequence in which the lysate may be processed.

The most extensively used procedures early on were differential centrifugation and acid precipitation for phage purification and concentration [73]. Two-phase separation made its appearance during the 60 s as an alternative concentration method, later reported on by Bachrach and Friedmann as they compared it to several purification and concentration methods for their time (namely centrifugation and precipitation).

Precipitation may be utilized in place of centrifugation for phage purification. Purification studies have been conducted which demonstrate the utility of precipitation compared to centrifugation for viral vectors such as bovine rotavirus and AAV [78]. In addition, rather than precipitating the virus itself, some researchers have instead precipitated host cell proteins and nucleic acids [75,79]. Like the commonly-used CsCl_2_ centrifugation method, however, precipitation may also reduce phage activity (measured in PFU/mL) [74]. Another potential drawback of precipitation was elucidated in the Kröber et al. study with influenza viruses, which found that the isoelectric point of the particles contributed to product losses by co-precipitation with cationic polymers [75,79]. Precipitation may be improved for utility with phages by enhancing process robustness and identifying nontoxic compounds to be applied in pharmaceutical production [75].

### 4.3. Clarified Lysate Treatments

Following the treatment of the crude lysate using a custom assortment of the previously described methods, the clarified lysate can then be processed using finer purification procedures, such as filtration and chromatography.

#### 4.3.1. Filtration

Phage filtration can be performed with a range of different pore sizes. 0.45 μm membranes are useful for removing uninfected or phage resistant bacteria. If the phage happens to aggregate, it may be retained by the filter as well [80]. Ultrafiltration is a common process for the concentration of therapeutic drug, such as antibodies, and for the removal of unwanted material, such as host cell proteins and DNA [81]. Similarly, this unit of operation can be applied for the concentration of virus and removal of cell fragments and damaged viruses [82,83]. Ultrafiltration can be used in the process of creating a phage drug product. One use for this unit of operation is for the concentration of phage before or after a purification step. UF has been added as part of a protocol for the making and preparation of phage for therapy [84]. Secondly, this method can be used for the removal of enterotoxins and endotoxins, which can be harmful to the human body, so it is important to get rid of them [85]. One group used this method with a 100 kDa MWCO PES filter and achieved a 20-fold reduction in enterotoxins from phage lysate [86]. Although the same group only saw a small reduction in endotoxins, this method can still provide benefit to a manufacturing phage process when paired with other unit of operations to make up for it lacks.

#### 4.3.2. Chromatography

The separation of phages from host cell proteins and other extraneous molecules is dependent on the interaction of the phage with the column stationary phase [75]. Wolf and Reichl outlined four main points for chromatography: physical structure and surface chemistry of the stationary phase, mobile phase composition, chromatography mode, and equipment. Focusing on these points in consideration of phage characteristics are expected to lead to a successful purification. The following chromatography methods are typically used for virus purification and may be adapted for a given process.

Ion-exchange chromatography for phages is dependent on their charge—since the head and tail are charged opposite of one another, one could use specific ions to bind either the head or the tail. Although cesium chloride gradient density centrifugation is commonly used for purification, anion-exchange chromatography can be used for this process as well [87]. Phage purification using anion-exchange chromatography has shown to provide recovery rates similar to those of using CsCl_2_ gradient density ultracentrifugation while being completed in a shorter period of time [88]. The recovery rates with this method could improve with proper protocol modifications for a specific phage, as different phages have been shown to require different adjustments in the protocol for this method [89]. The time being saved in this process can save equate to money being saved in a manufacturing setting. This method has been shown to work with 10+ different phages and can provide a higher titer yield at larger scale compared to the CsCl_2_ method, and because of the reusability of a column, long term costs can be reduced with the reuse of a same column [76,90]. Recently, a large-scale anion-exchange column has been developed and manufactured by BiaSeparations that can be used for phage capture and has been used in a pilot plant study for phage purification at a larger than bench scale level [91]. The recovery rates of phages with this method has been reported to be 10%–30%, indicating that modifications can be made to the method for improvements and higher rates [92]. Although those rates are low, higher rates have been reported for this column at a similar scale with recovery rates of 55% when loaded with virus for the purpose of purification [93]. This column type has also been used for purification of multiple different types of viruses [87,94]. An anion-exchange chromatography protocol has been written for phage purification with the intention of being an alternative for CsCl_2_ density gradient purification of phages and to be used at multiple scales [90,95,96].

Monolithic chromatography may also serve as an alternative for resin-based methods. Owing to phage size, they can potentially get trapped in the void volumes (or within resin particles) in traditional chromatography methods. Monolithic chromatography is made up of a single porous monolith which can be modified to incorporate other types of chromatography, such as anion exchange or ligand binding. The pores in the monolith can be charged or contain ligands which can bind the phage and make it easier to isolate them [89].

## 5. Discussion

As is evident from the breadth of considerations, the choice of phage and host has massive implications on the manufacturability, and indeed the success of the potential drug product at commercial scale. We urge the reader to consider manufacturing challenges and opportunities throughput the earliest stages of bacteriophage product development. Manufacturing and related activities are to be understood as running in parallel to research activities, not in sequence.

## Figures and Tables

**Figure 1 pharmaceuticals-14-00934-f001:**
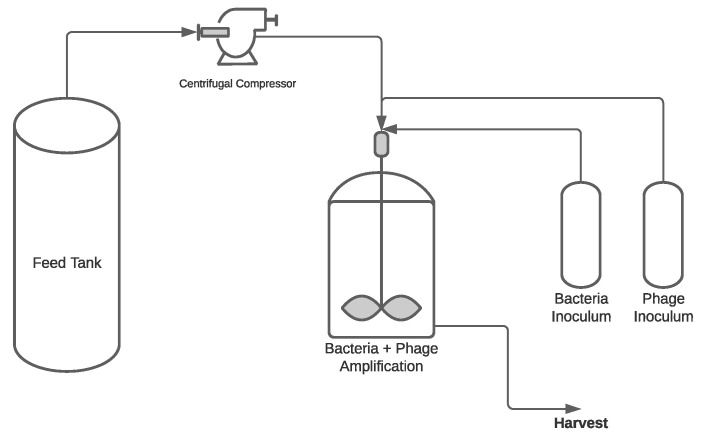
A simplified process flow diagram for phage production in batch operation mode.

**Figure 2 pharmaceuticals-14-00934-f002:**
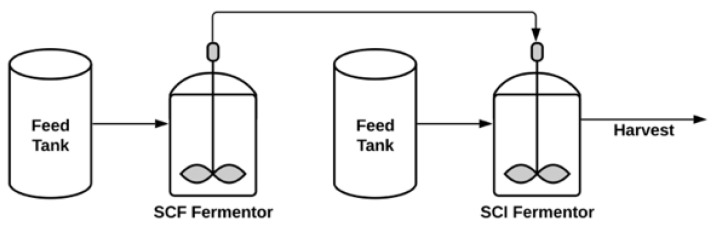
A simplified process flow diagram of the proposed SCF/SCI system where SCF and SCI are fermenters for bacteria and phage amplification, respectively. All the control systems are excluded to provide a simple depiction of the system. Adapted from [69].

**Figure 3 pharmaceuticals-14-00934-f003:**
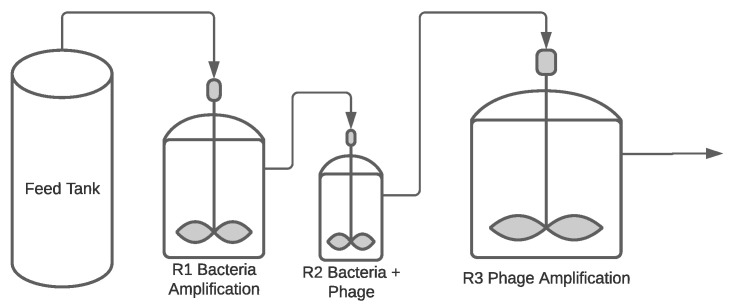
A simplified PFD of a continuous phage manufacturing system. As before, all the control systems are excluded to provide a simple depiction of the system.

**Figure 4 pharmaceuticals-14-00934-f004:**
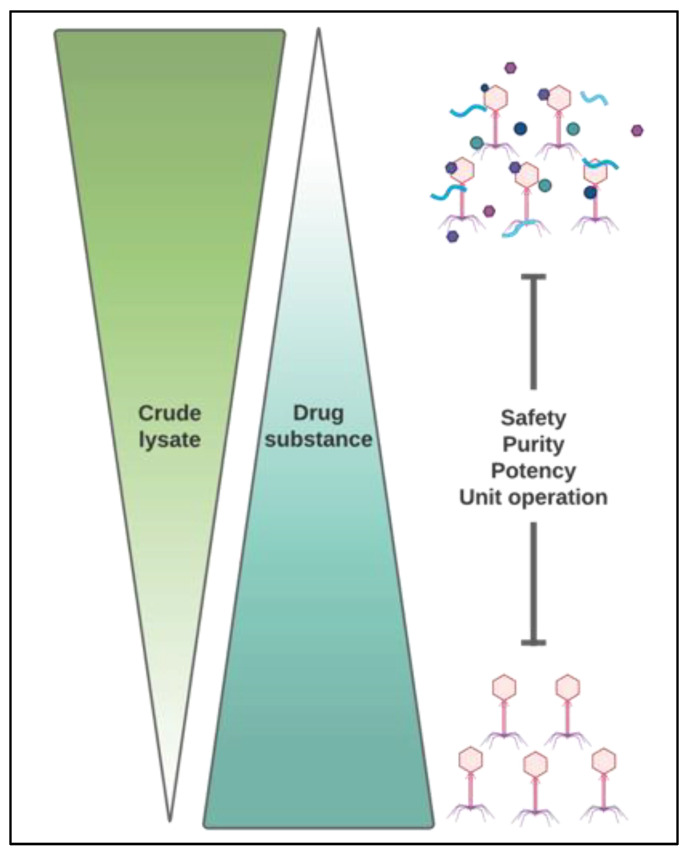
It is important to realize that crude lysate starts becoming DS and needs to be treated as such. The process developer should consider the critical quality attributes that are measured along the downstream separation cascade.

**Figure 5 pharmaceuticals-14-00934-f005:**
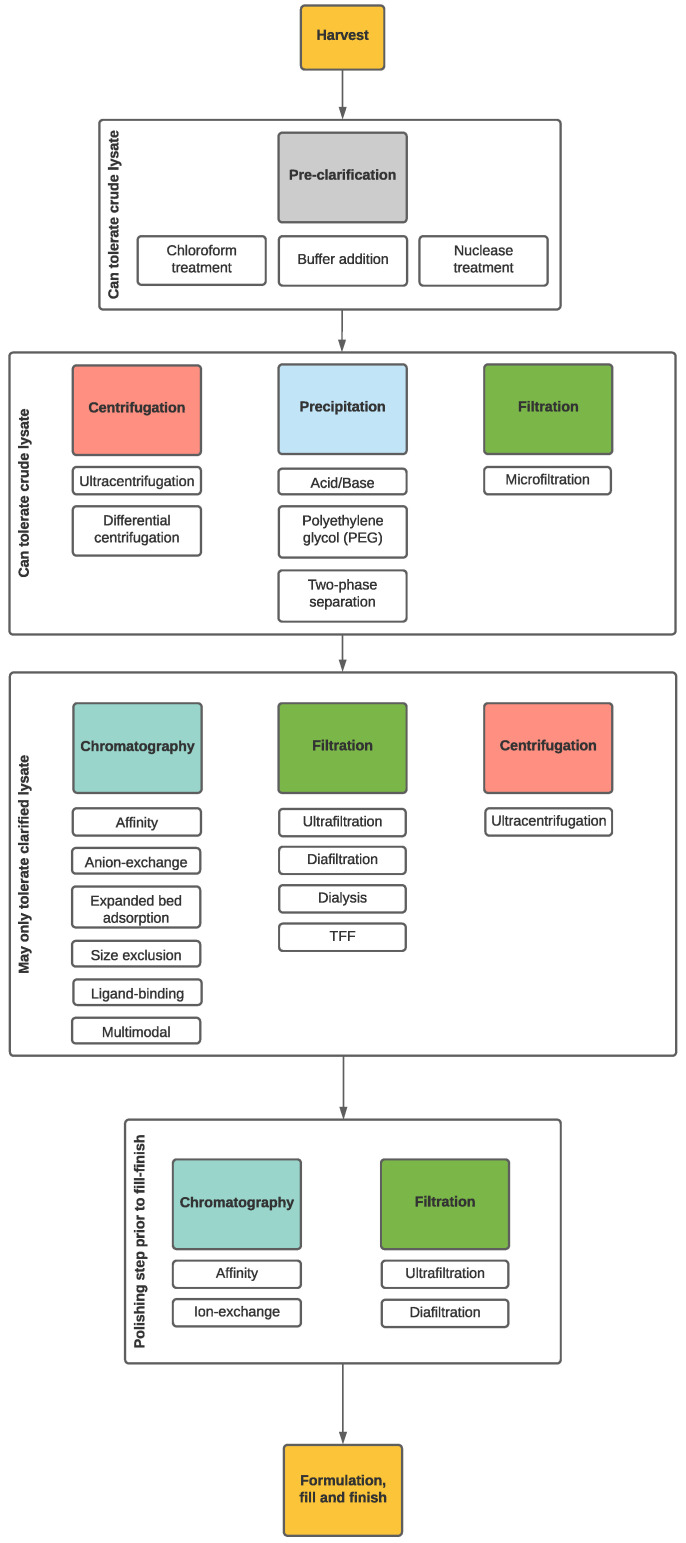
Generalized process flow for phage purification. From harvest, pretreatment is required to collect the phages from the lysate. Crude lysate can be processed using a combination of centrifugation, microfiltration, and precipitation. Once the lysate has been concentrated and the host cell debris reduced, more sensitive procedures may be implemented, such as chromatography and specific types of filtrations. Polishing procedures may be required to further isolate and purify the phage before moving on to fill-finish.

**Table 1 pharmaceuticals-14-00934-t001:** Overview of phage production modes.

Operation Mode	Advantages	Disadvantages	References
Batch	High titers (from 5 × 10^12^ PFU/mL to 1 × 10^16^ PFU/mL) Cheapest to produce phage	Conditions change during the process Long preparation time Reduction in overall productivity Limited by the maximum equipment volume available Potential batch to batch variation	Jurač, Nabergoj and Podgornik, 2018 [65]; Warner et al., 2014 [66]; Sochocka et al., 2015; Nabergoj et al., 2018 [9]
Semi-Continuous	In processes where bacteria are grown separately from phages, phage-resistance is avoided.	Self-cycling system requires advanced and integrated control and monitoring	Mancuso, Shi and Malik, 2018 [67]
Continuous	Higher overall productivity Cost savings Consistent and higher quality product is obtained due to easier control Operational complexity is reduced	Laborious because the system is complex A totally continuous process may lead generation of phage-resistant strains if the required measures are not taken Requires close monitoring to sustain steady-state Expensive to implement the system	Mancuso. Shi and Malik, 2018 [67]

## Data Availability

Data sharing not applicable.

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
