# Peer review of "Manufacturing Bacteriophages (Part 1 of 2): Cell Line Development, Upstream, and Downstream Considerations"

_pharmaceuticals, 2021, doi:10.3390/ph14090934_

Round 1
Reviewer 1 Report
This article aims to give an overview of the process leading to bacteriophages as drug substance. The authors covered a range of aspects from cell line development to upstream and downstream processing.
Such review is very helpful to understand the developments of phage-related drug substance. However, it will be nice if the authors can summary the potential regulatory of FDA and EMA associated with phage products. And it is not clear what kinds of criterion will be applied for cell line development in section 2.1. Furthermore, sections 2.2 and 2.3 do not make much sense to me. Once phages and hosts have been selected, it is for sure that other phages and host contamination are not allowed otherwise they cannot be used as seeds for production. Section 2.3 is just like talking about the phage titration. I would suggest the authors reorganizing sections 2.1-2.3 as two sections to summarize the selections of phages and production hosts respectively.
Minor comments
- Lines 47-48: please take care of “[INSERT REF WHEN ASSIGNED]”
- Line 494: “sAs” should be “As”?
- The format of references has to be corrected.
Author Response
This article aims to give an overview of the process leading to bacteriophages as drug substance. The authors covered a range of aspects from cell line development to upstream and downstream processing.
Such review is very helpful to understand the developments of phage-related drug substance.
- However, it will be nice if the authors can summary the potential regulatory of FDA and EMA associated with phage products.
- Authors response: We agree with Reviewer 1 and will insert language that points the interested reader to a further review in this special issue. This review will compare and contrast the FDA and EMA stance, as well as take into account the recent NIAID/FDA Bacteriophage Workshop (8/30-9/2) findings and discussions.
- And it is not clear what kinds of criterion will be applied for cell line development in section 2.1.
- Authors response: We appreciate the reviewers comments and the echo other reviewers comments. We clarified and streamlined section 2.1, along with re-titling it to ” Manufacturing Inclusion Criteria for Candidate Phages”. The section is included in this comment for ease of reference:
- During the initial selection, specific attention must be paid to upstream yield to satisfy inclusion criteria. It is critical to know how many phages are needed for an efficacious therapy, yet the preference from the process development (PD) and CLD side is for minimizing the total number of phages, and the total number of production hosts needed, while maintaining clinical efficacy. The combination of phage and production host further drives choices as yield discrepancies between phage-host combinations within the cocktail are a significant risk for PD success and overall manufacturing. To be clear, it is critical to know the potency of the phage DS, which on the upstream side is initially a product of lysis time and burst size. For these types of experiments, different phage concentrations should be tested on a given host concentration and lysis timing should be measured. A measure of phage fitness can be described as the ratio of phages to bacteria, or multiplicity of infection (MOI) [4]. The time that the highest ratio is achieved would be considered the maximum fitness of that phage for the given concentration tested in liquid media [5]. Although it seems that the maximum fitness can be achieved from a greater lysis time due to phages having more time to replicate in a host, the opposite can be true. In the presence of a high host concentration, lysis time is shortened as phages are allowed to infect more host cells and replicate more frequently [6]. Consequently, the shorter lysis time and greater infection rate leads to a higher phage to bacteria ratio; a better phage fitness is achieved and host concentration must be calculated to provide the best burst size for the phage [6]. The shortened lysis time is due to the available resources in the environment that allow for a larger phage population [7]. Another factor to consider before introducing a phage to the host is the host density. Some phages only show replication at a certain host density, so it is important to know the density required before phages are introduced in a bioreactor [8]. Besides host density, host growth rate also plays a role in phage lysis time and burst size. It has been shown that bacteria with a higher growth rate induce a higher burst size in phages and host replication at the maximum growth rate induces the highest burst size [7], [9]. Therefore, after determination of the host density for growth, the maximum growth rate can be determined if a large burst size is desired.
- Furthermore, sections 2.2 and 2.3 do not make much sense to me. Once phages and hosts have been selected, it is for sure that other phages and host contamination are not allowed otherwise they cannot be used as seeds for production.
- Authors response: We believe the Reviewer refers primarily to section 2.2. (Purity and Identity) in their comments vis-à-vis contamination. It is clear contamination is unacceptable, however we point out the need to rigorously characterize the potential for prophage contamination through a manufacturing host. This is an effort that is aided by bioinformatic methods such as we described but is not exhaustive. Therefore, it is crucial to utilize microbiological, chemical and physical methods to determine the prophage generation potential for a particular manufacturing cell line under conditions that are not just at or about the likely operating parameters for the upstream manufacture but go above and beyond that.
An additional consideration is that for engineered bacteriophages the stability of the engineering has to be continuously verified and that means that for the master phage bank and working phage bank the identity requirements are stringent and a “contamination” is not necessarily an external contaminant but a drifting species that split from the main lineage throughout propagation.
- Authors response: We believe the Reviewer refers primarily to section 2.2. (Purity and Identity) in their comments vis-à-vis contamination. It is clear contamination is unacceptable, however we point out the need to rigorously characterize the potential for prophage contamination through a manufacturing host. This is an effort that is aided by bioinformatic methods such as we described but is not exhaustive. Therefore, it is crucial to utilize microbiological, chemical and physical methods to determine the prophage generation potential for a particular manufacturing cell line under conditions that are not just at or about the likely operating parameters for the upstream manufacture but go above and beyond that.
- Section 2.3 is just like talking about the phage titration. I would suggest the authors reorganizing sections 2.1-2.3 as two sections to summarize the selections of phages and production hosts respectively.
- Authors response: We appreciate the Reviewers comments but urge them to contemplate the section as a standalone module. Assessing the potency of the master phage bank is a critical consideration that historically has been approached using various microbial techniques such as titration into a double-agar-overlay assay. However, we raise the concern that the potency should – indeed must – be validated in a number of different ways, as it is the authors opinion that there is no one definitive potency assay for bacteriophages. The citations in this module should direct the interested reader towards the relevant resources to educate themselves further on the pros and cons of various methods.
- Authors response: We appreciate the reviewers comments and the echo other reviewers comments. We clarified and streamlined section 2.1, along with re-titling it to ” Manufacturing Inclusion Criteria for Candidate Phages”. The section is included in this comment for ease of reference:
Minor comments
- Lines 47-48: please take care of “[INSERT REF WHEN ASSIGNED]”
- Revised.
- Line 494: “sAs” should be “As”?
- Revised.
- The format of references has to be corrected.
- Agreed, will revise with editorial staff.
Reviewer 2 Report
Dear authors,
The article is rather interesting for manufacturers, however I would suggest to pay more attention to selection of phages and development of phage cocktails.
Here are my comments, some of them minor, but other require careful attention.
In the authors’ list the second affiliation of the corresponding author “Michael S. Koeris 1,2,*” is not mentioned.
Line 23: Please, insert an abbreviation for “drug substance” which is further used as DS in lines 29, 30, etc.
Line 47: [INSERT REF WHEN ASSIGNED] – What does this mean? When do you plan to insert the reference? NOTE: Once the reference is inserted, the whole list of references and the citation numbers in the text numbers may change.
Lines 60-63: A measure of phage fitness can be described as the ratio of phages to bacteria, and the time that the highest ratio is achieved would be considered the maximum fitness of that phage for that given concentration tested [4].
Line 60: “The ratio of phage to bacteria” is a multiplicity of infection (MOI), is not it? So, why don’t you use this term?
Further in the text (lines 77-79) you mention that host bacteria density is important for phage replication. Is not this the same as MOI? My impression is that more clarifications is required. This part echoes with the part 3.2.3. Multiplicity of Infection (MOI).
When you are saying: “the time that the highest ratio is achieved would be considered the maximum fitness of that phage for that given concentration tested” (Lines 62-63), I believe you mean that the phages in this case are propagated in liquid media. In my opinion this fact should be clarified.
The text in lines 63-65 and lines 69-71 are duplicated, please compare:
Lines 63- 65: Although it seems that the maximum fitness can be achieved from a greater lysis time due to phages having more time to replicate in a host, the opposite can be true. In the presence of a high host concentration, lysis time is shortened and phages are allowed to infect more host and replicate more.
Lines 69 -71: Although it seems that the maximum fitness is achieved from a greater lysis time due to phages having more time to replicate in a host, the opposite can be true. In the presence of a high host concentration, lysis time is shortened and phages are allowed to infect more host and replicate at a greater rate [6].
A difference between the two texts above is highlighted.
Similarly, the texts in lines 66-68 and lines 73-75 are duplicated, please, compare:
Lines 66-68: Consequently, the shorter lysis time and greater infection rate leads to a higher phage to bacteria ratio, so a better phage fitness is achieved and host concentration must be calculated to provide the best burst size of phage [5].
Lines 73-75: Consequently, the shorter lysis time and greater infection rate leads to a higher phage to bacteria ratio, so a better phage fitness is achieved and host concentration must be calculated to provide the best burst size of phage [5].
Altogether the description of the part 2.1. Selection is rather vague and required careful reading and serious improvement.
Lines 104-117, part 2.3. Potency.
If the authors are using the term potency for phages considering them as drug substances (DS) or drug products (DP), this means that they understand the term “potency” as explained by Weatherall (The meaning an importance of drug potency in medicine, 1966, Clinical Pharmacology and Therapeutics, 7(5): 77-582. https://doi.org/10.1002/cpt196675577). Here is a citation: ”The term “potency” is ambiguous, because it may refer to the quantity of drug necessary to produce a given effect, or it may refer to the maximum response which can be achieved with the drug….. Potency cannot be measured in precise and absolute terms without detailed specification of the method of measurement, and then it becomes an excessively arbitrary measurement. Relative potency is more easily determined, but it also is variable according to the circumstances in which it is assessed. An approximate estimate of potency is afforded by the dose ordinarily used therapeutically.” The measurement method described in the part 2.3. Potency only partly corresponds to the above explanation, however it does not reflect therapeutic effect of phages. Phage enumeration aiming determination of the phage titer (i.e. a number of infective particles in ml of volume), which is usually done on a single host or a limited number of the potential hosts, does not demonstrate therapeutic potency of phages. In my opinion the measurement of phage potency should be based on a host range (or host spectrum or coverage), which seems to be more important rather than the titer. The broader is the host range the better therapeutic effect may be expected. Besides, different phage components when mixed in the cocktail, can decrease their titers not only because they will be diluted, but because of the physiological processes occurring in such cocktails. In this case a fitness of each phage does not matter much, the main point here is an ability of different phage clones to coexist in a mixture and develop an efficient and balanced system. Although the term Potency is very modern and sounds pharmaceutically, it does not seem to correspond to phage enumeration method as described in this part. I would suggest to rename this part and change it to phage titer. Also, it would be reasonable to add more information on development of phage mixtures that are erroneously termed as phage cocktails.
Line 122: Long term instead of song-term
Line 158: Should be considered instead of should be consider
Line 262: Delete “in” in the end of line.
Lines 272-273: Mutation rate should be “10-3” in the negative power. By the way, the mentioned mutation rate is determined for hosts bacteria or phages. This is not clear. According to Sanjuán et al. (2010) a mutation rate for dsDNA bacteriophages is 1–8 × 10−7 mutations per nucleotide per infection.
Legend to Fig.2. It would be easier to understand if you would explain that SCF fermentor is for bacterial propagation and SCI fermentor is for propagation of phages.
Line 470: Should be: ”…phages have been shown to require….”
Line 484: Improve CsCl2 to CsCl2
Line 494: Delete “s” in the beginning of the sentence “sAs s is evident from the breadth of considerations…..”
References: Put an interval between the citation number and the authors initials. See, numbers from 1 through 9.
Example: [1]L. J. Schiff, “Production, Characterization, and Testing of Banked Mammalian Cell Substrates Used to Produce Bio- 506 logical Products,” In Vitro Cell. Dev. Biol. Anim., vol. 41, no. 3/4, pp. 65–70, 2005.
Author Response
The article is rather interesting for manufacturers, however I would suggest to pay more attention to selection of phages and development of phage cocktails.
Here are my comments, some of them minor, but other require careful attention.
- In the authors’ list the second affiliation of the corresponding author “Michael S. Koeris1,2,*” is not mentioned.
- Author’s response: Thank you for your comment. The second affiliation for Michael S. Koeris has been removed
- Line 23: Please, insert an abbreviation for “drug substance” which is further used as DS in lines 29, 30, etc.
- Author’s response: Abbreviation inserted for “drug substance”. Line 23
- Line 47: [INSERT REF WHEN ASSIGNED] – What does this mean? When do you plan to insert the reference? NOTE: Once the reference is inserted, the whole list of references and the citation numbers in the text numbers may change.
- Author’s response: Thank you. Issue Resolved. Line 49
- Lines 60-63: A measure of phage fitness can be described as the ratio of phages to bacteria, and the time that the highest ratio is achieved would be considered the maximum fitness of that phage for that given concentration tested [4].
- Author’s response: We agree this section needed clarification and have expanded on the linear relationship between host lysis time and burst size relative to the density of the total bacterial population. Lines 68-77.
- Line 60: “The ratio of phage to bacteria” is a multiplicity of infection (MOI), is not it? So, why don’t you use this term?
- Author’s response: We agree that in one form of the ratio the multiplicity of infection is a useful expression, however we suggest that MOI as it has been historically understood generally refers to the initiation of an infectious process. For a continuum of ratios that are changing throughout the process of growing bacteria (clarified in line 67-68) and generation progeny phage, we believe the more general term of the phage to bacteria ratio is simpler for the reader and does not lead them down a particular path of thought. Line 64-68
- Further in the text (lines 77-79) you mention that host bacteria density is important for phage replication. Is not this the same as MOI? My impression is that more clarifications is required. This part echoes with the part 3.2.3. Multiplicity of Infection (MOI).
- Author’s Response: We agree that clarification was necessary and have highlighted the differences. Density is different as it relates to host population per volume that is independent of the phage. Resolved. Line: 76-83
- When you are saying: “the time that the highest ratio is achieved would be considered the maximum fitness of that phage for that given concentration tested” (Lines 62-63), I believe you mean that the phages in this case are propagated in liquid media. In my opinion this fact should be clarified.
- Author’s response: We agree and have made this distinction more apparent in lines 63-65
- The text in lines 63-65 and lines 69-71 are duplicated, please compare:
- Lines 63- 65: Although it seems that the maximum fitness can be achieved from a greater lysis time due to phages having more time to replicate in a host, the opposite can be true. In the presence of a high host concentration, lysis time is shortened and phages are allowed to infect more host and replicate more.
Lines 69 -71: Although it seems that the maximum fitness is achieved from a greater lysis time due to phages having more time to replicate in a host, the opposite can be true. In the presence of a high host concentration, lysis time is shortened and phages are allowed to infect more host and replicateat a greater rate [6].
A difference between the two texts above is highlighted.- Author’s response: Thank you for pointing out the duplication, duplicate lines were removed.
- Similarly, the texts in lines 66-68 and lines 73-75 are duplicated, please, compare:
Lines 66-68: Consequently, the shorter lysis time and greater infection rate leads to a higher phage to bacteria ratio, so a better phage fitness is achieved and host concentration must be calculated to provide the best burst size of phage [5].
Lines 73-75: Consequently, the shorter lysis time and greater infection rate leads to higher phage to bacteria ratio, so a better phage fitness is achieved and host concentration must be calculated to provide the best burst size of phage [5].- Author’s response: The duplicate lines 73-75 have been removed.
- Altogether the description of the part1. Selection is rather vague and required careful reading and serious improvement.
- Author’s response: Thank you for your comment. We clarified the title of section 2.1 to “Manufacturing Inclusion Criteria for Candidate Phages” to be specific and responsive to this first section on inclusion criteria. The second half of the entry was removed because it was duplicative of prior statements. Line 52
- Lines 104-117, part 2.3. Potency.
If the authors are using the term potency for phages considering them as drug substances (DS) or drug products (DP), this means that they understand the term “potency” as explained by Weatherall (The meaning an importance of drug potency in medicine, 1966, Clinical Pharmacology and Therapeutics, 7(5): 77-582. https://doi.org/10.1002/cpt196675577). Here is a citation: ”The term “potency” is ambiguous, because it may refer to the quantity of drug necessary to produce a given effect, or it may refer to the maximum response which can be achieved with the drug….. Potency cannot be measured in precise and absolute terms without detailed specification of the method of measurement, and then it becomes an excessively arbitrary measurement. Relative potency is more easily determined, but it also is variable according to the circumstances in which it is assessed. An approximate estimate of potency is afforded by the dose ordinarily used therapeutically.”
The measurement method described in the part3. Potency only partly corresponds to the above explanation; however it does not reflect therapeutic effect of phages. Phage enumeration aiming determination of the phage titer (i.e. a number of infective particles in ml of volume), which is usually done on a single host or a limited number of the potential hosts, does not demonstrate therapeutic potency of phages. In my opinion the measurement of phage potency should be based on a host range (or host spectrum or coverage), which seems to be more important rather than the titer. The broader is the host range the better therapeutic effect may be expected. Besides, different phage components when mixed in the cocktail, can decrease their titers not only because they will be diluted, but because of the physiological processes occurring in such cocktails. In this case a fitness of each phage does not matter much, the main point here is an ability of different phage clones to coexist in a mixture and develop an efficient and balanced system. Although the term Potency is very modern and sounds pharmaceutically, it does not seem to correspond to phage enumeration method as described in this part. I would suggest renaming this part and change it to phage titer. Also, it would be reasonable to add more information on development of phage mixtures that are erroneously termed as phage cocktails.- Author’s response: Thank you for your comment. For clarification purposes we adjusted the title to “Master Phage Bank Design”. For cell line development, the considerations we lay out are necessary to be taken into account. Line 105-121.
- Line 122:Long term instead of song-term
- Author’s response: Thank you. That was Resolved, Line 125
- Line 158:Should be considered instead of should be consider
- Author’s response: Thank you. That was resolved, Line 162
- Line 262: Delete “in” in the end of line.
- Author’s response: Thank you. That was resolved., Line 266
- Lines 272-273: Mutation rate should be “10-3” in the negative power. By the way, the mentioned mutation rate is determined for hosts bacteria or phages. This is not clear. According to Sanjuán et al. (2010) a mutation rate for dsDNA bacteriophages is 1–8 × 10−7mutations per nucleotide per infection.
- Author’s response: Thank you for pointing this out! It is an important concept and should be clarified to the readers. In that context, we aimed to emphasize that phages have higher mutation rates compared to their hosts. That means mutations may accumulate to a point that would lead the final product to undergo a significant change. That would cause the therapeutic fail to provide the aimed function. Therefore, we aimed to point out to the interested researchers to consider this fact while they are designing their processes to produce phage therapeutics. Resolved, Line 273-278
- Legend to Fig.2. It would be easier to understand if you would explain that SCF fermentor is for bacterial propagation and SCI fermentor is for propagation of phages.
- Author’s response: Thank you for your comment! The clarification for the SCF/SCI system is added to the figure description. Resolved, Line 313-318
- Line 470: Should be: ”…phages have beenshown to require….”
- Author’s response: Thank you. This error has been Resolved, Line 475
- Line 484: ImproveCsCl2 to CsCl2
- Author’s response: We appreciate the suggestion. It has been Resolved, Line 489
- Line 494: Delete “s” in the beginning of the sentence “sAs s is evident from the breadth of considerations…..”
- Author’s response: Thank you for pointing that out. It has been Resolved, Line 500
- References: Put an interval between the citation number and the authors initials. See, numbers from 1 through 9.
Example:[1] L. J. Schiff, “Production, Characterization, and Testing of Banked Mammalian Cell Substrates Used to Produce Bio- 506 logical Products,” In Vitro Cell. Dev. Biol. Anim., vol. 41, no. 3/4, pp. 65–70, 2005.- Author’s response: The formatting of a “space” has been inserted. Resolved throughout. Furthermore, we have been in contact with the editorial staff and the references will be further formatted by them.

Reviewer 3 Report
It is an interesting approach in a world of antibiotic resistance. Being only the first part, it is not very sure what the result is and how the work will be composed for completion.
Work with bacteriophages occurs around 1900, has been forgotten for a while, re-appears. There is significant potential in the use of bacteriophages.
Please review and correct typing errors. For example, writing degrees Celsius is wrong every time. I wouldn't write western blotings. There are other bibliographic resources including in Central and Eastern Europe. Multi-centric collaboration is important for a successful project.
Author Response
- It is an interesting approach in a world of antibiotic resistance. Being only the first part, it is not very sure what the result is and how the work will be composed for completion.
- Authors response: this is the first part of the two-part series on a comprehensive review on manufacturing considerations to produce either engineered or unmodified bacteriophage drug product from cell line development all the way through to formulation considerations. We urge the reviewer and all interested parties to consider the two reviews in a joint context (https://www.mdpi.com/1424-8247/14/9/895/pdf).
- Work with bacteriophages occurs around 1900, has been forgotten for a while, re-appears. There is significant potential in the use of bacteriophages.
- Authors response: we wholeheartedly agree with the Reviewer on the potential for bacteriophage therapy especially in the context of recent resurgence in funding, development and progress in the field. Furthermore, we hope that this contribution allows the interested drug developer to gain a quick education in the field and find footing to begin or continue the development of critical and life-saving medicines.
- Please review and correct typing errors. For example, writing degrees Celsius is wrong every time. I wouldn't write western blotting’s. There are other bibliographic resources including in Central and Eastern Europe. Multi-centric collaboration is important for a successful project.
- Authors response: We thank the Reviewer for the close read of the manuscript, all the necessary adjustments to the representation of degrees Celsius have been made and the minor grammatical mistake have been corrected. We further agree with the Reviewer that the excellent work of especially researchers in Central and Eastern Europe is deserving of recognition and repetition within the advancement of the field.